# Toward New Assessment in Sarcoma Identification and Grading Using Artificial Intelligence Techniques

**DOI:** 10.3390/diagnostics15131694

**Published:** 2025-07-02

**Authors:** Arnar Evgení Gunnarsson, Simona Correra, Carol Teixidó Sánchez, Marco Recenti, Halldór Jónsson, Paolo Gargiulo

**Affiliations:** 1Institute of Biomedical and Neural Engineering, Reykjavik University, 102 Reykjavik, Iceland; carols@ru.is (C.T.S.); marcor@ru.is (M.R.); paolo@ru.is or paologar@landspitali.is (P.G.); 2Department of Medicine and Health Sciences “Vincenzo Tiberio”, University of Molise, 86100 Campobasso, Italy; s.correra@studenti.unimol.it; 3Department of Science, Landspitali University Hospital, 105 Reykjavik, Iceland

**Keywords:** sarcoma, radiomics, image transform, machine learning, classification

## Abstract

**Background/Objectives:** Sarcomas are a rare and heterogeneous group of malignant tumors, which makes early detection and grading particularly challenging. Diagnosis traditionally relies on expert visual interpretation of histopathological biopsies and radiological imaging, processes that can be time-consuming, subjective and susceptible to inter-observer variability. **Methods:** In this study, we aim to explore the potential of artificial intelligence (AI), specifically radiomics and machine learning (ML), to support sarcoma diagnosis and grading based on MRI scans. We extracted quantitative features from both raw and wavelet-transformed images, including first-order statistics and texture descriptors such as the gray-level co-occurrence matrix (GLCM), gray-level size-zone matrix (GLSZM), gray-level run-length matrix (GLRLM), and neighboring gray tone difference matrix (NGTDM). These features were used to train ML models for two tasks: binary classification of healthy vs. pathological tissue and prognostic grading of sarcomas based on the French FNCLCC system. **Results:** The binary classification achieved an accuracy of 76.02% using a combination of features from both raw and transformed images. FNCLCC grade classification reached an accuracy of 57.6% under the same conditions. Specifically, wavelet transforms of raw images boosted classification accuracy, hinting at the large potential that image transforms can add to these tasks. **Conclusions:** Our findings highlight the value of combining multiple radiomic features and demonstrate that wavelet transforms significantly enhance classification performance. By outlining the potential of AI-based approaches in sarcoma diagnostics, this work seeks to promote the development of decision support systems that could assist clinicians.

## 1. Introduction

Sarcomas are rare malignant tumors that develop in connective tissues such as bones, muscles, fat and cartilage. Due to their aggressive nature and histological diversity, sarcomas present significant clinical challenges in both diagnosis and treatment. Despite advances in oncology, accurately predicting the development and progression of sarcomas remains a major obstacle for surgeons and clinicians [1].

One primary challenge is diagnosing sarcomas, due to their variable clinical presentations. These tumors can appear in diverse and often nonspecific forms, leading to frequent delays in diagnosis and subsequent therapeutic intervention [2,3]. Due to the rarity of sarcoma tumors, accurate diagnosis is often delayed, resulting in patients suffering a higher mortality rate. Therefore, developing methods for early detection and diagnosis is crucial to improve the quality of life and survivability of patients.

When diagnosing a sarcoma tumor, the practitioner determines a grade representing the severity of the cancer. This scale has three grades: 1, 2 and 3, with each grade increasing in severity. This grading is a critical factor for prognosis and treatment decisions and still largely relies on traditional histopathological evaluation methods [4]. Imaging exams play a crucial role in the diagnostic process for sarcomas. Conventional X-rays and CT scans are essential for evaluating bone sarcomas, while ultrasound and MRI are key for assessing soft tissue sarcomas. These imaging techniques form the foundation of research, providing essential data for accurate diagnosis and classification of these complex tumors.

Although histopathological evaluation is the standard practice, it has significant limitations. The subjectivity inherent in interpretation of results by pathologists, combined with the complex characteristics of sarcomas, can lead to variations in diagnosis. Nearly 70% of sarcomas are larger than 5 cm at the time of diagnosis, underscoring the impact of diagnostic delays [2]. Furthermore, the similar morphological features of certain sarcoma subtypes increase the risk of human error.

To address these challenges, genetic diagnosis has emerged as a promising solution to enhance the accuracy and speed of sarcoma detection [5]. Genetic analysis can reveal specific mutations and molecular characteristics often missed by traditional methods, offering a more personalized approach to treatment. However, despite its potential, genetic diagnosis alone can struggle with the complex heterogeneity of sarcomas and may not always provide enough information for accurate malignancy grading or real-time decision-making [6]. This is where artificial intelligence (AI) comes in as a powerful complementary tool. While genetic data identifies key molecular drivers, AI can analyze imaging data to predict malignancy grades and detect subtle patterns missed by the human eye, bridging the gap where genetic analysis may fall short. Together, AI and genetic diagnosis have the potential to revolutionize sarcoma management, improving both the accuracy and speed of diagnosis and ultimately enhancing patient outcomes [7]. Given these challenges faced by both radiologists and pathologists, the integration of artificial intelligence (AI) into the diagnostic and classification process for sarcomas could offer innovative solutions. AI has the potential to standardize and significantly improve the accuracy of sarcoma diagnoses by reducing evaluation variability and providing supportive tools for these specialties. In fact, AI is already being implemented in various areas of healthcare, particularly in diagnostics, where it assists in interpreting medical images such as CT scans, MRIs and X-rays with remarkable accuracy [8,9]. Additionally, AI is being used in the detection of diabetic retinopathy [10] and neurological conditions like Alzheimer’s disease [11], enabling earlier intervention and improving patient outcomes.

The overall aim of this study was to analyze different radiomic features and the role they play in classifying healthy and pathological tissue, as well as sarcoma severity. This was done by training a random forest (RF) model to first classify healthy and pathological tissue, followed by training it to classify severity. This research aimed to demonstrate the feasibility and effectiveness of AI in supporting the diagnosis of sarcomas, highlighting the potential for integrating these technologies into routine clinical practice. We hope to contribute to improved patient management by providing an innovative tool that is complementary to existing diagnostic methods.

## 2. Methods

The workflow for this paper is outlined in Figure 1 and highlights the main steps. First, the raw magnetic resonance imaging (MRI) data are segmented into 3 masks—sarcoma, edema and healthy tissue—to produce masks. Then, simple linear iterative clustering (SLIC) segmentation is applied to the masks to generate super pixels (SPs). This is followed by radiomics feature extraction of each SP to generate a table of feature values, which are applied to raw image SPs and wavelet-transformed SPs. A feature selection step is applied separately for the different feature transform groups, and remaining features are finally used for the classification of healthy and pathological tissue (binary classification) and the severity or the grade of the sarcoma (grade classification) by applying a random forest (RF) model. Relevant metrics such as accuracy, precision, recall and confusion matrices are calculated, along with feature importance, to assess and analyze the results.

### 2.1. Data

This study used the IceSG (Icelandic Sarcoma Group) dataset, a collection of sarcoma cases assembled by Landspítali University Hospital and that is not publicly accessible. The use of the patient data was reviewed by the Landspítali Ethics Committee (dated 6 February 2024) and the Landspitali Scientific Research Committee (dated 9 January 2024) and approved.

For the current work, 101 patients’ MRI scans are considered for their ability to capture soft tissue details. If a patient has scans from multiple perspectives (axial, coronal, and/or sagittal), they are all included. Both T1 and T2 weighted scans are considered, with a mix of contrast media and with fat suppression both on and off. All sarcoma subtypes are considered. Out of the patient cohort, 19 are diagnosed with grade 1, 16 with grade 2 and 66 with grade 3. For each patient, the severity of prognosis is available at the time of the scan according to the French classification (FNCLCC) [12], which assigns a grade from 1 to 3 to the sarcoma, with 3 being the most severe.

Table 1 displays general demographics among age groups of the dataset, including sarcoma subtypes, FNCLCC grade and sex distributions.

### 2.2. Segmentation

Segmentation is the process of marking regions of an image into a group to create a mask. In this study, the sarcoma tumor was segmented, as was the surrounding edema (where available), both of which were labeled as pathological tissue. The remaining tissue was segmented and labeled as non-pathological tissue; therefore, the healthy dataset was retrieved from the sarcoma patients. The orientation of scans was not standardized during data gathering; therefore, all orientations are considered, including axial, coronal and sagittal scans. All orientations available for each patient were segmented, so there are up to 3 image sets segmented per patient.

Mimics (Materialize: Leuven, Belgium [13]) was used as the segmentation software. All segmentations were performed by hand and reviewed by a specialist. After segmentation, the raw images and resulting masks were exported for further processing.

### 2.3. Data Pre-Processing

After segmentation, 2946 images contained at least some trace of sarcoma and/or edema. There were 566 available images of grade 1 sarcoma, 395 of grade 2 and 1916 of grade 3. Due to the relatively small size of the dataset, a sampling technique was introduced to expand it for a purpose to that described by Karim Hammoudi et al. [14].

Simple linear iterative clustering (SLIC) is an iterative and algorithmic method that splits an image into a user-defined number of segments based on similarity measures in pixel intensity [15]. Using this method, it is possible to further split the regions of interest into more data points for further processing, similar to Z. Khatun et al. [16], who segmented Achilles tendons using SLIC. In this manner, smaller details are captured, and the issue of a small dataset is somewhat addressed. The newly generated segments are referred to as super pixels (SPs). The target size of SPs is set to 500 pixels. This size should offer a good balance between capturing smaller and larger details. It should be noted that due to the nature of the SLIC algorithm, the actual number of pixels in each segment may vary. This research considers image slices individually; therefore, all generated super pixels are 2-dimensional.

The sampling method was applied separately for the pathological region and the non-pathological region. The technique generated approximately 40,000 pathological and 820,000 non-pathological data points, with roughly 5900 grade 1 data points, 3600 for grade 2 and 30,000 for grade 3. The approximate ratio of data points per patient is 312, 228 and 457 for grades 1, 2 and 3, respectively.

### 2.4. Feature Extraction

To acquire any meaningful information from the images and segmented regions, radiomics were employed to extract relevant features. Namely, the Python (version 3.9) PyRadiomics package [17] was applied for its vast set of extractable features that contain multiple feature categories. Feature groups include first-order statistics describing the distribution of pixel intensity (e.g., mean, variance and skewness). Gray-Level Co-occurrence Matrix (GLCM) features calculate features based on gray-level intensities and intensity relationships through the comparison of pixel pairs. Gray-Level Size-Zone Matrix (GLSZM) features are similar to GLCM features but compare regional intensities instead of pair-wise relationships. Gray-Level Run-Length Matrix (GLRLM) features measure how pixels of similar intensity group together. Neighboring Gray Tone Difference Matrix (NGTDM) features focus on differences between pixels and their neighbors, quantifying details such as contrast. Shape features are omitted from this study, as only 2-dimensional SPs are generated; therefore, only 2-dimensional features can be extracted, whereas shape requires 3-dimensional image data.

Wavelet functions are often used to extract higher order features from images. They decompose an image into different components representing different frequency bands and have been shown to increase classification performance [18]. It is possible to apply a low-pass and a high-pass wavelet filter in each dimension (X and Y in this case) along each image, which creates new feature classes. The same features are extracted from the original raw image and the transformed images. Therefore, with the 4 combinations of wavelet transforms, the number of extracted features increases by five times. Table 2 shows the types of wavelet transforms and the details they capture. Features extracted from the raw image are referred to as original features, and those extracted from the wavelet transforms are referred to as wavelet features.

All in all, a total of 465 features were extracted and analyzed for their classification ability (see Table 3).

### 2.5. Feature Selection

The feature selection step is important to increase model robustness and computational efficiency. Given a large number of features, some may not correlate with the feature being predicted and can introduce noise, making classification more difficult. The chosen model for this study is the random forest (RF) classifier, as it is generally robust, accurate and scalable and has an internal feature-ranking mechanism that computes the importance of every feature [20].

As RFs are not very sensitive to highly correlated and low-variance features, the maximum allowable correlation between features is set to 0.99, and only a single correlated feature of a set is used. Once highly correlated features have been filtered out, the model is trained on the rest of the features and on the entire dataset to acquire a feature ranking.

To find the best set of features, a 10-fold cross validation is applied, splitting the dataset into 10 groups patient-wise, using 9 for training and 1 for testing. Features are eliminated, from lowest importance to highest, until an optimal model has been trained based on the average area under the receiver operating curve (AUC-ROC) score across all 10 folds. The AUC-ROC score is a measure of how well the model discriminates between the predicted classes or how well the classes are separated. The set of features that results give the best AUC-ROC score is used for subsequent training.

### 2.6. Binary Classification

After data pre-processing, the dataset is split into 2 groups: healthy and pathological, with the aim of analyzing which radiomic features best distinguish between healthy and pathological tissue. The healthy cohort of data points is randomly selected from the the SPs generated from the healthy mask component of the diagnosed patients. Each patient contributes an equal number of pathological and healthy data points. In this manner, a balanced dataset for training and testing is ensured.

A leave-1-out cross-validation using an RF classifier is applied such that the training dataset is maximized and as much variance as possible is captured by it and explained by the model.

Three types of analyses are conducted, using different feature combinations, to explore the effects that they have on the overall results. The first run includes only raw image (original) features. The second considers only wavelet-transformed features, omitting the LL wavelet transform, as it is an approximation of the raw image. The third run combines original features with the wavelet features. Refer to Table 3 for the combination of image transform spaces for each run. Feature selection is applied independently on each combination of image transforms, selecting the only features used for the three binary classifiers; therefore, the number of optimal features can differ.

The accuracy, precision, recall and the confusion matrix (see Figure 2a for confusion matrix description) are the metrics calculated to asses each feature set, and feature importance is calculated to analyze their effects in classification. Averaged macro and micro metrics are also calculated. Macro metrics give equal weight to each patient during calculations, and micro metrics give equal weight to each data point during calculations. The macro and micro confusion matrices are expressed as follows:(1)$$C_{Ma}=\frac{1}{N_{p}}\sum_{i=1}^{N_{p}}\frac{C_{i}}{\sum_{j,k}c_{i,j,k}}$$(2)$$C_{Mi}=\sum_{i}^{p}C_{i}$$
where $N_{p}$ is the total number of patients, $C_{i}$ is the confusion matrix of the *i*th patient and $c_{i,j,k}$ denotes the value at line *j* and row *k* of the *i*th patient’s confusion matrix.

### 2.7. Grade Classification

Following the FNCLCC guidelines, there are 3 potential severities of sarcoma prognosis: grades 1, 2 and 3. These are based off of tumor differentiation, mitotic count and tumor necrosis. As the aim is to extract radiomic data from the patient MRI scans to analyze which features might explain sarcoma severity, only the grade is applied for prediction purposes, along with extracted radiomics data.

The dataset comprises an imbalanced cohort of patients: 19 patients receiving a grade 1 diagnosis, 16 with grade 2 and 66 patients with grade 3 sarcoma. This issue is tackled using a balanced random forest model [21] such that the models do not overly adhere to one specific class.

As for binary classification, three analyses are compiled (see Table 3). The first only considers original features; the second only considers wavelet transforms, excluding the LL wavelet transform; and the third is a combination of the feature sets. Again, feature selection is applied independently on each combination of image transforms, so the number of features can differ for each transform combination trial. Macro and micro accuracy, precision, recall and the confusion matrix (see Figure 2b for confusion matrix description) are calculated and used for assessment, as is the feature importance. In this case, macro-averaged calculations depend on how the patient was classified (based on majority rules), and micro-averaged calculations depend on how each individual segment was classified.

The micro confusion matrix is calculated using Equation (Equation 2), and the macro confusion matrix is calculated as follows:(3)$$C_{Ma}=\sum_{i}^{N_{p}}C_{i}^{\prime }$$
where(4)$$C_{i}^{\prime }:c_{i,j,k}^{\prime }=\left\{\begin{array}{cc} 1, & c_{i,j,k}=max\left(C_{i}\right)\\ 0, & \mathrm{otherwise} \end{array}\right.$$
where subscript i,j,k denotes the values of the *j*th row and *k*th column in the *i*th patient’s confusion matrix.

## 3. Results

The presented results are based on the leave-one-out cross-validation applied to the images processed from the IceSG dataset. After SLIC was applied to each image to generate super pixels (SPs), features were extracted from the raw image, and the four wavelet transform passes. A random forest classifier was applied first to the raw (original) images, then to the wavelet-transformed images and, finally, to the combination of the two to first distinguish between healthy and pathological tissue (binary classification), then to classify the severity or grade of the prognosis. Macro- and micro-averaged accuracy, precision, recall and confusion matrices, along with feature importance, were calculated and used to quantify each run.

The resulting accuracy, precision and recall for the different combinations of features show a general improvement when using more features. First-order statistical and GLCM features were generally the most important, and GLSZM and NGTDM were generally the lowest ranked feature groups. No single transform feature group performed significantly better throughout all runs, with importance varying depending on the situation.

### 3.1. Binary Classification: Healthy–Pathological

Table 4 displays the overall results of the binary classification results. The macro- and micro-averaged accuracy, precision and recall are displayed for original, wavelet and combined feature groups.

#### 3.1.1. Original Features

When considering only original features, macro-averaged accuracy, precision and recall of 72.66%, 76.80% and 68.80% were obtained, respectively. The micro-averaged accuracy, precision and recall were 68.70%, 72.14% and 67.79%, respectively. The following confusion matrices were obtained:
Macro Confusion Matrix:$\left[\begin{array}{cc} 0.3826 & 0.1174\\ 0.1560 & 0.3440 \end{array}\right]$Micro Confusion Matrix:$\left[\begin{array}{cc} 27,686 & 12,088\\ 12,810 & 26,964 \end{array}\right]$

Table 5 displays the number of features chosen from each feature group, and Table 6 displays the cumulative importance for each feature group. A total of 47 features were selected, with all feature groups being represented. First order ranked as the most important, followed by GLCM, GLDM, GLRLM, GLSZM and NGTDM.

#### 3.1.2. Wavelet Features

Considering pure wavelet features, omitting the LL wavelet pass of the transform resulted in a macro-averaged accuracy, precision and recall of 70.16%, 72.75% and 66.68%, respectively. The micro-averaged accuracy, precision and recall were 64.79%, 66.86% and 62.27%, respectively. The following confusion matrices were obtained:
Macro Confusion Matrix:$\left[\begin{array}{cc} 0.3682 & 0.1318\\ 0.1666 & 0.3334 \end{array}\right]$Micro Confusion Matrix:$\left[\begin{array}{cc} 26,773 & 13,001\\ 15,008 & 24,766 \end{array}\right]$

Table 7 displays the number of features chosen from each feature group, and Table 8 displays the cumulative importance for each feature group. A total of 71 features were selected, with nearly all feature groups being represented for each wavelet transform. The highest ranking feature group in terms of importance was first order, followed by GLCM, GLDM, GLRLM, GLSZM and NGTDM. The wavelet-HL transform was the most important, followed by wavelet-LH and and wavelet-HH transforms.

#### 3.1.3. Combined Features

Combining the wavelet transform feature groups with the original feature group boosted performance and resulted in a macro-averaged accuracy, precision and recall of 76.02%, 80.88% and 71.82%, respectively. The micro-averaged accuracy, precision and recall were 71.88%, 75.82% and 68.05%, respectively. The following confusion matrices were obtained:
Macro Confusion Matrix:$\left[\begin{array}{cc} 0.4011 & 0.0989\\ 0.1409 & 0.3591 \end{array}\right]$Micro Confusion Matrix:$\left[\begin{array}{cc} 30,114 & 9660\\ 12,706 & 27,068 \end{array}\right]$

Table 9 displays the cumulative importance for each feature group, and Table 10 displays the number of features chosen from each feature group. A total of 200 features were selected, with all but one transform feature group being represented for each transform group. The highest ranking feature group in terms of importance was first order, followed by GLCM, GLRLM, GLDM, GLSZM and NGTDM. The original transform group was the most important, followed by wavelet-LL, wavelet-HL, wavelet-HH and wavelet-LH.

### 3.2. Grade Classification: Sarcoma Grade

#### 3.2.1. Original Features

Table 11 displays the results of grade classification using only original features. The table implies a higher difficulty in classifying grades 1 and 2 when compared to grade 3. Overall accuracy shows that the model is able to classify above chance (33%), but precision and recall for grades 1 and 2 could use a lot of improvement.

The following confusion matrices were obtained for the macro- and micro-averaged classification results:
Macro Confusion Matrix:$\left[\begin{array}{ccc} 7 & 5 & 20\\ 1 & 5 & 12\\ 11 & 6 & 34 \end{array}\right]$Micro Confusion Matrix:$\left[\begin{array}{ccc} 1685 & 768 & 7829\\ 1407 & 1644 & 7068\\ 2843 & 1244 & 15,286 \end{array}\right]$

The number of chosen features and feature importance can be seen in Table 12 and Table 13, respectively. Unsurprisingly, first-order features ranked the highest in importance, followed by GLSZM, NGTDM, GLCM, GLDM and GLRLM.

#### 3.2.2. Wavelet Features

Table 14 displays the results of grade classification, applying only wavelet features. The results are quite similar to the original feature run; there were general improvements to the metrics, with a slight reduction in macro accuracy, grade 3 precision, grade 2 recall and micro grade 2 recall.

The following macro- and micro-averaged confusion matrices were obtained:
Macro Confusion Matrix:$\left[\begin{array}{ccc} 8 & 1 & 18\\ 4 & 9 & 21\\ 7 & 6 & 27 \end{array}\right]$Micro Confusion Matrix:$\left[\begin{array}{ccc} 2066 & 749 & 6551\\ 1736 & 2045 & 6551\\ 2133 & 862 & 17,081 \end{array}\right]$

The number of selected features and the feature importance for each wavelet transform type can be seen in Table 15 and Table 16, respectively. The tables indicate that the wavelet-HH transform was most important, followed by wavelet-HL and wavelet-LH. GLCM features ranked as most important, followed by GLRLM, first order, GLDM, NGTDM and GLSZM.

#### 3.2.3. Combined Features

Using the combined feature set of both original and wavelet features showed some general improvements (see Table 17). Macro-averaged accuracy is slightly lower when compared to the pure wavelet transform, although the micro-averaged is at the highest of all runs. Grade 3 macro- and micro-averaged precision and recall were relatively high compared to the other runs, and grade 2 precision was quite high, while the other metrics suffered.

The resulting confusion matrices are
Macro Confusion Matrix:$\left[\begin{array}{ccc} 6 & 3 & 16\\ 0 & 5 & 16\\ 13 & 8 & 34 \end{array}\right]$Micro Confusion Matrix:$\left[\begin{array}{ccc} 1935 & 576 & 6577\\ 1142 & 2201 & 4831\\ 2858 & 879 & 18,775 \end{array}\right]$

Table 18 and Table 19 show the number of selected features and the feature importance, respectively. The highest ranking feature group was first order, followed by GLCM, GLRLM, NGTDM, GLDM and GLSZM. Wavelet-HH was the most important transform group, followed by the original, wavelet-LL, wavelet-HL and, finally, wavelet-LH.

## 4. Discussion

The results of this study highlight the potential of radiomics for sarcoma classification. Tagliafico et al. [22] found that GLCM features were key in distinguishing healthy tissue from local recurrence (LR) in soft tissue sarcoma (STS), a finding consistent with this paper’s results. They also identified four features correlated with tumor grade. Yu Zhang et al. [23] used a standardized MRI protocol on 33 patients to successfully differentiate between low- and high-grade STS. Similarly, Corino et al. [24] showed that first-order statistics were the most important features in distinguishing between high- and intermediate-grade sarcomas. While few studies have applied wavelet transforms for sarcoma classification, Shan Hu et al. [25] found that Sym4 and Db4 wavelet transforms combined with GLCM could accurately detect sarcoma. Peng Lin et al. [26] applied 3D wavelet transforms for preoperative osteosarcoma (OS) evaluation with positive results, and Silin Chen et al. [27] combined raw and wavelet features for STS prognosis prediction. Niu et al. [28] applied various image transformations to MRI glioma volumes to predict tumor grade with high accuracy. Radiomics-based predictive modeling has also demonstrated use cases beyond oncology. Ricciardi et al. [29] applied tree-based algorithms (including random forests and gradient boosting) to radiodensitometric parameters obtained from mid-thigh CT scans. They achieved high performance for coronary heart disease, cardiovascular disease and chronic heart failure, supporting the relevance of radiomic features for clinical risk stratification.

The feature importance analysis aligns with existing literature [22,24], confirming the utility of wavelet transforms to improve random forest (RF) model performance [25,26]. Most studies focus on a single sarcoma subtype with standardized imaging, which improves model generalization. However, there are several limitations in this study: dataset complexity, the use of image transform algorithms, dataset imbalance and limitations associated with clinical sarcoma grading.

### 4.1. Dataset Complexity

The dataset is complex, as sarcomas are diverse, making generalization difficult [8]. Variations in MRI scan protocols, such as differing contrast, fat suppression and weighting methods, contribute to variability in tumor appearance (Figure 3). Considering multiple orientations—axial, coronal and sagittal—adds complexity that the model must handle. Additionally, differences in gray-value ranges and MRI specifications complicate standardization. Some sarcoma types, like OS, are under-represented, making grade classification challenging, as the model may misclassify OS as a lower grade. Including sarcoma subtype information could improve classification.

Non-pathological tissue was grouped as healthy, which can cause misclassification, as biological tissue may resemble pathological regions (Figure 4). Small homogeneous regions in segmented 3D images may confuse the model, especially when similar to healthy tissue. Image quality varied significantly between patients (Figure 5), with some images being grainy, blurred or corrupted, which may affect model performance.

To improve generalization, it is crucial to balance the dataset by ensuring adequate representation of all sarcoma types and grades. For grade classification, the recommended number of samples per feature is at least 10 [30], with a maximum of 39 features for three-grade classification based on the dataset’s smallest minority-class size when considering each slice of the mask as a data point. Hence, SLIC was used to cluster pixels into super pixels for further analysis.

### 4.2. Image Transform Algorithms

Wavelet transforms were used for feature extraction, but other transforms, such as Hessian transform [31], could provide additional relevant information. Applying different transforms, such as Laplacian of Gaussian or gradient transforms [17], may uncover additional textural or topological features that could improve classification performance. However, caution is needed when using higher order transformations, as they may amplify image noise [32].

While combining original and transformed features improved performance for binary classification, there was no single feature that consistently predicted outcomes for all patients, underscoring the complexity of feature interactions and suggesting that optimal feature selection could improve results. Using the combined dataset did not improve performance to a notable degree for grade classification, which was unexpected. This implies that improvements must be made to the feature selection or, perhaps, indicating the need to include more transforms.

### 4.3. Dataset Imbalance

To balance healthy and pathological groups, healthy data were randomly selected per patient to match the number of pathological data points. However, this process could lead to slight variations, especially in patients with small tumors. Over-sampling could address this issue, improving generalization during training. Despite dataset challenges, the classification results were generally successful, highlighting the interdependence of features. The findings indicate that there is considerable potential in identifying sarcoma tissue based on radiomics features.

The under-representation of grade 1 and 2 tumors further complicates the task, as grade 3 tumors, being larger, generate more SPs [33]. Precision and recall were poorer for grades 1 and 2, as seen in Table 11, Table 14 and Table 17.

To address class imbalance, a balanced random forest model was used [21], but other techniques, such as random oversampling or ensemble methods, could also be effective [20,21]. The results suggest that improvements can be made in grade classification, particularly for grades 1 and 2, though the low number of cases in these categories should be considered.

However, the overall results offer a possible pathway of applying computerized models to accurately diagnose patients. This could decrease the time taken to diagnose patients, as no invasive biopsies are required and another practitioner does not have to analyze the results of the biopsy.

### 4.4. Clinical Sarcoma Grading

Sarcoma grading, according to FNCLCC guidelines, relies on tumor differentiation, the necrosis ratio and the mitotic count. These factors are subject to human error, making grade classification particularly challenging and leading to potential misdiagnosis of the sarcoma grade.

## 5. Conclusions

The results of training an RF classifier on the IceSG dataset to discriminate healthy versus pathological tissue and classify the sarcoma grade using radiomics features extracted from raw images and image wavelet transforms show considerable promise. The results indicate the following:A pathway may exist to high-accuracy identification of sarcomas and their grades.Medical imaging shows promise in bypassing invasive biopsies.Possible human error in complicated diagnostics can be minimized or avoided.Treatment of sarcoma can be started earlier, thereby promoting patient prognosis.

## Figures and Tables

**Figure 1 diagnostics-15-01694-f001:**
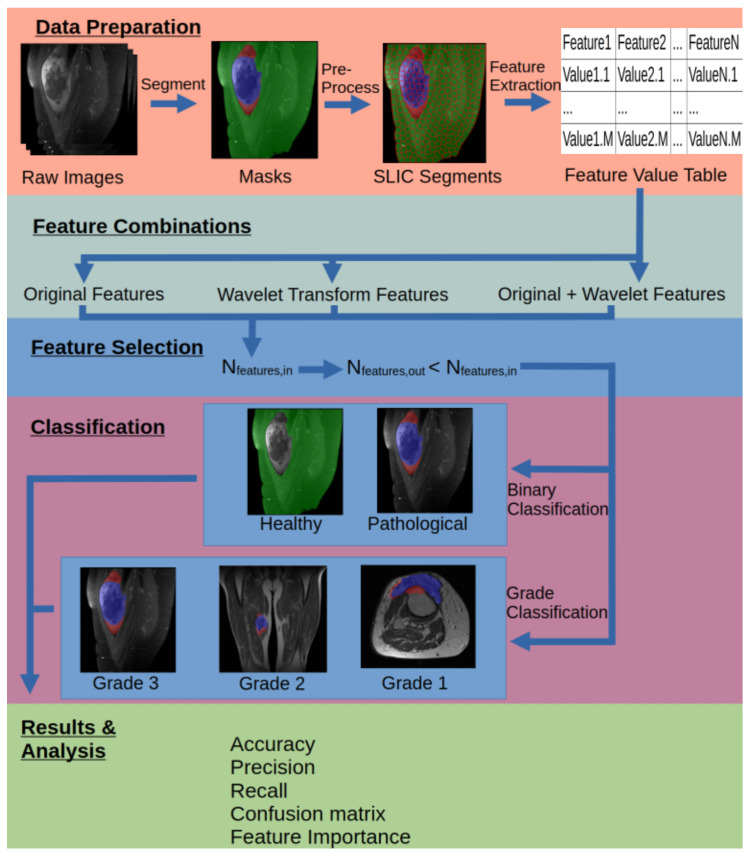
Visual representation of the steps in the workflow.

**Figure 2 diagnostics-15-01694-f002:**
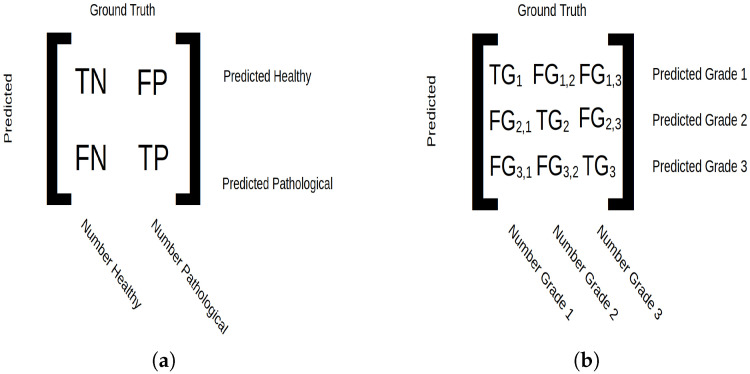
Confusion matrix setup. (**a**) For binary classification, true negative (TN) indicates tissue correctly predicted as healthy, true positive (TP) indicates tissue correctly predicted as pathological, false negative (FN) indicates pathological tissue predicted as healthy and false positive (FP) indicates pathological tissue predicted as healthy. The micro average displays the number of samples classified in integers, and macro average displays classification in ratios. (**b**) For grade classification, true grade ($TG_{i}$) is the number of samples belonging to class *i* and predicted as such, and false grade ($FG_{i,j}$) corresponds to samples predicted as grade *i* but belonging to grade *j*. The micro average shows the total number of classified data points, and the macro average shows number of classified patients.

**Figure 3 diagnostics-15-01694-f003:**
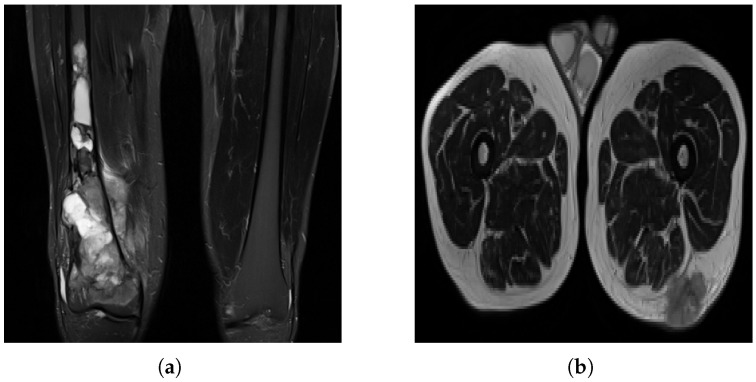
MRI slices: (**a**) A case of OS where the machine uses specific settings and contrast media to brighten the tumor region and suppress fat response. (**b**) Another patient (with liposarcoma in this case) where the tumor region is dark in contrast and fat tissue is not suppressed.

**Figure 4 diagnostics-15-01694-f004:**
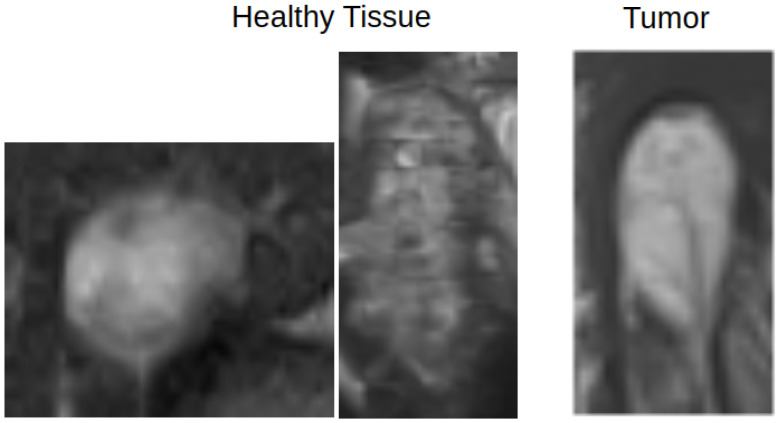
An example of how healthy and pathological tissue can overlap in terms of appearance. The left two images show healthy tissue, and the rightmost image shows a tumor region.

**Figure 5 diagnostics-15-01694-f005:**
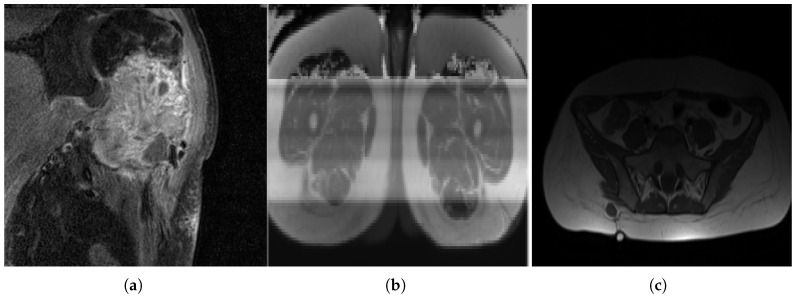
Variance in image quality. (**a**) A grainy slice with relatively high noise. (**b**) A corrupted image. (**c**) An example of uneven lighting distribution.

**Table 1 diagnostics-15-01694-t001:** Dataset demographics. Included sarcoma subtypes are chondrosarcoma (CS), dermatofibrosarcoma protuberans (DFSP), Ewing sarcoma (ES), giant cell tumor of bone (GCTB), osteosarcoma (OS) and soft tissue sarcoma (STS).

Age Group	0–19		20–39		40–59		60–79		80+	
Subtype										
STS	4	3.96%	10	9.90%	19	18.81%	22	21.78%	10	9.90%
OS	10	9.90%	0	0.00%	3	2.97%	1	0.99%	0	0.00%
CS	0	0.00%	1	0.99%	6	5.94%	0	0.00%	1	0.99%
ES	5	4.95%	4	3.96%	1	0.99%	1	0.99%	0	0.00%
DFSP	0	0.00%	1	0.99%	0	0.00%	0	0.00%	0	0.00%
GCTB	1	0.99%	0	0.00%	1	0.99%	0	0.00%	0	0.00%
Grade										
1	3	2.97%	0	0.00%	10	9.90%	3	2.97%	3	2.97%
2	1	0.99%	2	1.98%	3	2.97%	8	7.92%	2	1.98%
3	16	15.84%	14	13.86%	17	16.83%	13	12.87%	6	5.94%
Sex										
male	12	11.88%	8	7.92%	17	16.83%	14	13.86%	8	7.92%
female	8	7.92%	8	7.92%	13	12.87%	10	9.90%	3	2.97%

**Table 2 diagnostics-15-01694-t002:** Types of wavelet sub-band passes and captured details [19].

Sub-Band Symbol	X-Axis	Y-Axis	Details
LL	Low-pass	Low-pass	Approximationof original image
LH	Low-pass	High-pass	Images’ horizontal details
HL	High-pass	Low-pass	Images’ vertical details
HH	High-pass	High-pass	Images’ diagonal details

**Table 3 diagnostics-15-01694-t003:** The combination of image transforms for each run.

	Image Transform Group	Description	Number of Features
1.	Original	Raw image features	93 features
2.	Wavelet	Wavelet-transformed features	279 features
3.	Wavelet + Original	Raw and wavelet-transformed features	465 features

**Table 4 diagnostics-15-01694-t004:** Binary classification results. Both macro-averaged and micro-averaged metrics are displayed.

	Macro-Averaged			Micro-Averaged		
	Accuracy	Precision	Recall	Accuracy	Precision	Recall
Original	72.66%	76.80%	68.80%	68.70%	72.14%	67.79%
Wavelet	70.16%	72.75%	66.68%	64.79%	66.86%	62.27%
Combined	76.02%	80.88%	71.82%	71.88%	75.82%	68.05%

**Table 5 diagnostics-15-01694-t005:** Number of features chosen for each feature group for the binary classifier for the original transform group.

Feature Group	Selected Features
First Order	14
GLCM	9
GLDM	8
GLRLM	6
GLSZM	5
NGTDM	5
Total	47

**Table 6 diagnostics-15-01694-t006:** Feature importance considering purely the original transform group for binary classification.

Feature Group	Original
First Order	35.60%
GLCM	16.75%
GLDM	15.95%
GLRLM	12.23%
GLSZM	9.58%
NGTDM	9.90%
Total	100.00%

**Table 7 diagnostics-15-01694-t007:** Number of features chosen for each feature group for the binary classifier for the wavelet transform group.

	Selected Features			
Feature Group	Wavelet-LH	Wavelet-HL	Wavelet-HH	Total
First Order	9	10	4	23
GLCM	9	5	5	19
GLDM	6	5	1	12
GLRLM	5	2	4	11
GLSZM	2	0	0	2
NGTDM	1	2	1	4
Total	32	24	15	71

**Table 8 diagnostics-15-01694-t008:** Feature importance considering only the wavelet transform group for binary classification.

	Feature-Transform Group Importance			
Feature Group	Wavelet-LH	Wavelet-HL	Wavelet-HH	Total
Firstorder	14.28%	16.08%	5.39%	35.76%
GLCM	12.12%	7.04%	7.06%	26.22%
GLDM	8.25%	6.65%	1.38%	16.28%
GLRLM	6.34%	2.78%	5.33%	14.45%
GLSZM	2.31%	0.00%	0.00%	2.31%
NGTDM	1.19%	2.50%	1.30%	4.99%
Total	44.49%	35.05%	20.46%	100.00%

**Table 9 diagnostics-15-01694-t009:** Feature importance considering both original and wavelet transform groups for binary classification.

	Feature Transform Group Importance					
	Original	Wavelet-LL	Wavelet-LH	Wavelet-HL	Wavelet-HH	Total
First Order	13.75%	7.85%	1.86%	2.96%	3.26%	29.67%
GLCM	4.00%	4.56%	3.51%	5.05%	5.35%	22.47%
GLDM	3.99%	3.48%	4.06%	2.38%	2.70%	16.61%
GLRLM	2.57%	3.58%	3.27%	2.83%	4.71%	16.96%
GLSZM	2.10%	2.67%	0.78%	0.80%	0.00%	6.34%
NGTDM	1.65%	2.18%	1.37%	1.50%	1.26%	7.95%
Total	28.06%	24.32%	14.84%	15.51%	17.26%	100.00%

**Table 10 diagnostics-15-01694-t010:** Number of features chosen from each feature group for the binary classifier of all transform groups.

	Selected Features					
	Original	Wavelet-LL	Wavelet-LH	Wavelet-HL	Wavelet-HH	Total
First Order	15	10	4	7	9	45
GLCM	10	9	6	12	12	49
GLDM	8	7	8	6	7	36
GLRLM	5	7	7	7	10	36
GLSZM	6	6	2	2	0	16
NGTDM	4	4	3	4	3	18
Total	48	43	30	38	41	200

**Table 11 diagnostics-15-01694-t011:** Macro- and micro-averaged results for grade classification using original features.

	Macro			Micro		
	Grade 1	Grade 2	Grade 3	Grade 1	Grade 2	Grade 3
Accuracy	45.54%			46.80%		
Precision	36.84%	31.25%	51.52%	28.39%	44.97%	50.64%
Recall	21.88%	27.78%	66.67%	21.88%	27.78%	66.67%

**Table 12 diagnostics-15-01694-t012:** Number of features selected when considering the original transform group for grade classification.

Feature Group	Selected Features
First Order	11
GLCM	3
GLDM	3
GLRLM	1
GLSZM	8
NGTDM	5
Total	31

**Table 13 diagnostics-15-01694-t013:** Feature importance considering purely the original feature groups for grade classification.

Feature Group	Feature Group Importance
Firstorder	37.81%
GLCM	8.94%
GLDM	8.88%
GLRLM	3.12%
GLSZM	22.82%
NGTDM	18.43%
Total	100.00%

**Table 14 diagnostics-15-01694-t014:** Macro- and micro-averaged results for grade classification using wavelet features.

	Macro			Micro		
	Grade 1	Grade 2	Grade 3	Grade 1	Grade 2	Grade 3
Accuracy	43.56%			53.28%		
Precision	42.11%	56.25%	40.91%	34.81%	55.94%	56.59%
Recall	29.63%	26.47%	67.50%	29.63%	26.47%	67.50%

**Table 15 diagnostics-15-01694-t015:** Number of features selected when considering the wavelet transform group for grade classification.

	Selected Features			
	Wavelet-LH	Wavelet-HL	Wavelet-HH	Total
First Order	0	2	3	5
GLCM	4	4	3	11
GLDM	0	0	4	4
GLRLM	1	1	4	6
GLSZM	0	0	0	0
NGTDM	0	1	0	1
Total	5	8	14	27

**Table 16 diagnostics-15-01694-t016:** Feature importance considering only wavelet transform feature groups for grade classification.

	Feature Transform Group Importance			
	Wavelet-LH	Wavelet-HL	Wavelet-HH	Total
First Order	0.00%	7.29%	12.36%	19.65%
GLCM	17.78%	18.00%	9.47%	45.25%
GLDM	0.00%	0.00%	11.81%	11.81%
GLRLM	3.34%	3.48%	13.02%	19.84%
GLSZM	0.00%	0.00%	0.00%	0.00%
NGTDM	0.00%	3.44%	0.00%	3.44%
Total	21.12%	32.22%	46.67%	100.00%

**Table 17 diagnostics-15-01694-t017:** Macro- and micro-averaged results for grade classification using all transform groups.

	Macro			Micro		
	Grade 1	Grade 2	Grade 3	Grade 1	Grade 2	Grade 3
Accuracy	44.55%			57.60%		
Precision	31.58%	31.25%	51.52%	32.60%	60.20%	62.20%
Recall	24.00%	23.81%	61.82%	24.00%	23.81%	61.82%

**Table 18 diagnostics-15-01694-t018:** Number of features selected when considering both original and wavelet transform groups for grade classification.

	Selected Features					
	Original	Wavelet-LL	Wavelet-LH	Wavelet-HL	Wavelet-HH	Total
First Order	10	5	0	0	1	16
GLCM	0	1	2	4	3	10
GLDM	0	0	0	0	4	4
GLRLM	1	2	0	3	6	12
GLSZM	1	1	0	0	0	2
NGTDM	2	2	1	0	0	5
Total	14	11	3	7	14	49

**Table 19 diagnostics-15-01694-t019:** Feature importance considering both the original and wavelet transform feature groups for grade classification.

	Feature Transform Group Importance					
	Original	Wavelet-LL	Wavelet-LH	Wavelet-HL	Wavelet-HH	Total
First Order	19.57%	11.05%	0.00%	0.00%	2.47%	33.08%
GLCM	0.00%	1.62%	5.91%	9.19%	5.93%	22.65%
GLDM	0.00%	0.00%	0.00%	0.00%	7.60%	7.60%
GLRLM	1.64%	2.62%	0.00%	6.16%	11.68%	22.10%
GLSZM	1.35%	1.36%	0.00%	0.00%	0.00%	2.71%
NGTDM	4.57%	4.31%	2.97%	0.00%	0.00%	11.86%
Total	27.13%	20.97%	8.88%	15.34%	27.68%	100.00%

## Data Availability

The original contributions presented in the study are included in the article, further inquiries can be directed to the corresponding author.

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
