# Peer review of "Toward New Assessment in Sarcoma Identification and Grading Using Artificial Intelligence Techniques"

_diagnostics, 2025, doi:10.3390/diagnostics15131694_

Round 1
Reviewer 1 Report
Comments and Suggestions for Authors
The study presents the effectiveness of artificial intelligence and machine learning algorithms in sarcoma diagnosis and grading. In the paper, binary classification was performed by manually extracting features from both raw and wavelet transformed images. In addition, grading of sarcomas based on the French FNCLCC system was also performed.
1. How did you determine the effectiveness of the extracted features on the classification accuracy? Some of the extracted features may not have an effect on the classification accuracy.
2. More meaningful results can be obtained by using an algorithm such as Recursive Feature Elimination to increase the accuracy. The authors can try using this algorithm.
3. What do the rows and columns of the micro and macro confusion matrices represent? It is unclear.
4. What should we understand from the red or blue areas in the feature correlation matrix? Figure 5 is too small.
5. Section 4.1 should be defined just before the method section.
6. Why was manual feature extraction applied instead of automatic feature extraction? Please elaborate in the discussion section.
Author Response
- How did you determine the effectiveness of the extracted features on the classification accuracy? Some of the extracted features may not have an effect on the classification accuracy.
- The methods were changed slightly to include a round of feature elimination, this removed most features and boosted classification accuracy in many cases.
- Effectiveness is gauged using feature importance ranking, built into the random forest algorithm.
- More meaningful results can be obtained by using an algorithm such as Recursive Feature Elimination to increase the accuracy. The authors can try using this algorithm.
- RFE was applied in an extra step.
- What do the rows and columns of the micro and macro confusion matrices represent? It is unclear.
- A figure has been added describing confusion matrices along with equations describing how they are calculated, along with some supporting text.
- What should we understand from the red or blue areas in the feature correlation matrix? Figure 5 is too small.
- The figure was removed since it was only there to describe the high correlation between features - which were selectively removed using RFE
- Section 4.1 should be defined just before the method section.
- The section was renamed to dataset complexity, as it was seen that discussed material belonged more in the discussion as it is a limitation, deduced from the results of the experiment.
- Why was manual feature extraction applied instead of automatic feature extraction? Please elaborate in the discussion section.
- We believe this is now answered due to the implementation of RFE. Please let us know if it is not sufficient.
Attached is the file containing the changes in red (additions) and blue (deletions)

Reviewer 2 Report
Comments and Suggestions for Authors
Congratulations to the authors for this extensive work about the utility and feasibility of the Artificial Intelligence to support sarcoma diagnosis and grading. The study is well structured and the methods are widely and accurately described. The analysis are clear and the researchers reported all the reasonable limits of the study. The proposed approach show promising results and its utility in clinical practice is suggested. Only minor revisions are requested before publication: in the line 5 "aaim" should be "aim"; in the table 1 it would be helpfull to include the percentage.
Author Response
The spelling errors were fixed, and percentages were added to table 1.
Thanks for the review!

Round 2
Reviewer 1 Report
Comments and Suggestions for Authors
I recommend that the article be accepted in its current form.